# Does the Degree of Mutualism between *Epichloë* Fungi and *Botanophila* Flies Depend upon the Reproductive Mode of the Fungi?

**DOI:** 10.3390/jof8121270

**Published:** 2022-12-01

**Authors:** Thomas L. Bultman, Marlena Lembicz, Adrian Leuchtmann

**Affiliations:** 1Biology Department, Hope College, Holland, MI 49423, USA; 2Department of Systematics and Environmental Botany, Adam Mickiewicz University in Poznań, Uniwersytetu Poznańskiego 6, 61-614 Poznan, Poland; 3Institute of Integrative Biology, ETH Zurich, CH-8092 Zurich, Switzerland

**Keywords:** ascomycota, diptera, fungal endophyte, reproductive mode

## Abstract

*Epichloë* (Ascomycota: Clavicipitaceae) fungi can form an intriguing interaction with *Botanophila* flies. The fungi live within above-ground shoots of grasses. Some species (type I) only reproduce sexually by forming stromata on all host culms (choke disease). Stromata produce haploid spores (spermatia) that fertilize stromata of opposite mating type to form dikaryotic cells. A second category of *Epichloë* species (type II) produces stromata on only some of the host culms; culms without choke produce flowers and seeds. These *Epichloë* can reproduce asexually by invading host seed, as well as sexually. Female *Botanophila* flies visit stromata for feeding and oviposition. Spermatia pass through the gut of *Botanophila* intact and viable. Flies can cross-fertilize the fungus during defecation after egg laying. Hence, we described the interaction as a mutualism similar to pollination. Yet, subsequent work by others and ourselves showed that visitation by *Botanophila* flies was not necessary for cross fertilization of *Epichloë.* We believe these contradictory results can be reconciled from an evolutionary perspective, if one takes into account the reproductive mode of the fungus. We explore a novel hypothesis to reconcile this contradiction, its predictions and discuss ways in which to test them.

## 1. Introduction

### 1.1. The Interaction

*Epichloë* fungi can form an intriguing interaction with *Botanophila* flies. The interaction was first noted in the 19th century [1] and confirmed in the 20th century [2,3,4]. These workers observed that adult flies visit fungal stromata to obtain food and lay eggs, and larvae eat part of the fungal stroma. *Epichloë* fungi are the only known food source for the larvae [5]; for this reason, researchers considered them specialized parasites of the fungi.

### 1.2. Scope of the Review

We briefly review the pertinent work on this study system dating from 1872 to the present, with emphasis on work over the past four decades. We identify a discrepancy in the published literature and present a novel hypothesis to explain it.

### 1.3. Natural History of Study System and Early Work

*Epichloë* spp. (Ascomycota: Clavicipitaceae) live within grasses. Typically, one fungal individual lives in one host, invading most if not all above ground shoots. Some species (type III) reproduce only asexually by invading host seed and growing up with the host seedling; this is called vertical transmission [6,7]. Other species (type I) form stromata on all host culms in the spring. Stroma formation stifles or chokes out flower production by the host (hence, stroma formation is also called “choke”). Stromata produce haploid conidial spores that fertilize other stromata of opposite mating type and fuse to form dikaryotic cells that give rise to asci and haploid ascospores within perithecia (fruiting bodies) that line the stromatal surface [8]. As this is occurring, female *Botanophila* flies visit stromata for feeding and oviposition. Fly eggs hatch in about four days and larvae begin to feed on developing perithecia. Eggs are usually white with distinct longitudinal ridges, however it is common for many to be yellow or gray and lack ridges. These tend also to be slightly smaller than white eggs and they presumably are nonviable, as they apparently do not hatch (pers. obs). A third category of *Epichloë* species (type II) produces stromata on only some of the host culms; culms without choke produce flowers and seeds. Thus, these *Epichloë* can reproduce asexually by invading host seed, as well as sexually, as described above.

Following the pioneering work of the Kohlmeyers [5] and earlier investigators, we showed that *Epichloë elymi* (infecting *Elymus virginicus* and *Elymus canadensis*) is heterothallic; that is, it is an obligate outcrosser, with two mating types in a population [9]. Thus, the conidial spores are more correctly called spermatia, as they function like gametes in the life cycle of the fungus. That discovery set the stage for investigating the possibility that *Botanophila* flies play a role in moving spermatia between fungi of opposite mating types. Evidence for this idea came from experiments in which we manipulated the presence/absence of flies.

When female flies captured from the field were allowed access to newly egressed stromata, perithecia began to form on stromata several days later [10]; stromata without flies failed to produce perithecia. We concluded that flies transfer spermatia while visiting stromata for egg laying. How flies actually transfer the tiny spermatia was investigated through direct observation in the laboratory.

We found that immediately following egg laying, flies drag the tip of their abdomen across the stroma surface as they walk the full length of the stroma several times, often in a spiral pattern [11]. These observations were complemented with experimental evidence. Application of fly feces, suspended in distilled water to unfertilized stromata resulted in perithecia formation at the exact places were feces had been applied. Thus, spermatia pass through the gut of the fly intact and viable. By visiting several stromata, a fly should accumulate spermatia of both mating types in its digestive tract and thereby effect cross-fertilization when defecating after egg laying [12]. Based on these results the association between *Epichloë* and *Botanophila* was described as a mutualism that Tom Bultman and coworkers argued is functionally similar to pollination [11].

## 2. Contradictory Findings

Following our work, Rao and Baumann [13] reported findings that contradicted the mutualism hypothesis. Working in Oregon (USA) with *Epichloë typhina* infecting the grass *Dactylis glomerata* (introduced from Europe) in cultivated seed production fields, they found that fly larvae consumed considerable amounts of the fungal stromata, but that presence of the fly was not necessary for cross-fertilization of the fungus. Similarly, more contradictory studies came from one of us (ML) working in Poland. Those studies showed that flies were not required for cross fertilization of *E. typhina* on *D. glomerata* [14] or *E. typhina* on *Puccinellia distans* [15].

## 3. Reconciling the Contradiction

A number of possible proximate-level (i.e., functional) explanations were raised to try to reconcile the contradictory results from Oregon and Poland. Because the sites in Oregon were commercial fields, the grasses and stromata were crowded together in close proximity to one another. It is possible that water splash or direct contact could have moved spermatia between stromata. Wind could also be a possible vector, although a test for this with *E. elymi* gave no support for this mechanism [16]. This is not surprising as stromata draw considerable water through the infected culm and into the stroma, only to evaporate from the stroma surface [17] resulting in evaporative cooling [18]. Hence, stromata are moist rather than dry and dehiscent [16,17]. It was also suggested that *E. typhina* might not be heterothallic and thus could be self-compatible. Yet, genetic analyses of *E. typhina* in Oregon showed that they all have two mating genotypes represented among individuals [19], consistent with a heterothallic mating system. Another possible explanation is that flies visiting stromata of *E. typhina* in Poland and Oregon often fed but did not lay eggs and while feeding they transferred some spermatia clinging to their mouthparts or other body parts. In this way, perithecia could form with no evidence that the fly had been there. Yet, another possibility is ascosporic fertilization in which ascospores released by perithecia serve as spermatia and fertilize stromata [19]. Cross fertilization would initially have to occur, perhaps through transfer by *Botanophila* or other animals, and then wind-transported ascospores produced from these fertilizations would initiate subsequent cross fertilizations. This scenario would require a lengthy time period during which stromata are produced and it is not clear that this is common among *Epichloë* species.

The best explanation to date for the contradictory results is that animals other than *Botanophila,* like slugs, are responsible for transferring a substantial number of spermatia. Spermatia transfer by slugs at study sites in Oregon has been tested experimentally, and the data support this hypothesis [19]. Furthermore, slugs were implicated in cross fertilization of *E. typhina* infecting *Poa trivialis* in northern France [20]. Yet, slug visitation of stromata is not limited to type I *Epichloë,* as slugs also visit type II *E. elymi* [21].

We believe these contradictory studies can be reconciled from an evolutionary perspective, if one takes into account the reproductive mode of the fungus. Type I (only sexual reproduction) *Epichloë* would benefit from less dependence on *Botanophila* flies than type II (sexual and asexual reproduction) *Epichloë*. Members of the first group will incur a higher risk of reproductive failure due to their close dependence on *Botanophila* vectors (if, for example, a suitable *Botanophila* species is absent from a given area). On the contrary, type II *Epichloë* can reproduce without the services of the fly (asexually, through infecting host seeds). Thus, they can “afford” to evolve towards stronger specialization with the vector. That is, they have a greater assurance of reproduction due to the ability to reproduce asexually, as well as sexually.

Yet, some further refinement to the hypothesis is necessary. That is because not all type II *Epichloë* are the same. In some, like *E. elymi*, fungal individuals (growing within an individual clump of host grass) produce stromata on some host culms and not others (and culms without choke will produce flowers and seed into which *Epichloë* invades). Others, like *E. typhina* in *P. distans*, may choke all culms of the individual host and thus there is no host seed to invade, or an infected clump can produce no choke and potentially all host seeds can be infected by the fungus. Thus, some type II *Epichloë* are type II at the individual level [hereafter referred to as type II(ind)]. These fungal individuals can reproduce sexually and asexually and thus should be expected to form stronger specialization with *Botanophila* (as stated above). Type II *Epichloë* that operate at the population level [some individuals are type I and some are type III (no choke) [hereafter referred to as type II(pop)], lack individuals that can produce choke and infected seeds on the same host. Type II(pop) *Epichloë* individuals should form weaker specialization with flies because they cannot reproduce both sexually and asexually through invading host seed.

This argument is similar to that made to (partially) explain the self-fertilization capability of some plant species that possess flowers with adaptations to attract very specialized pollinators. For example, Darwin [22], p. 58 suggested some *Ophrys* orchids self-fertilize even though they have highly specialized pollinators due to the reduced seed set that can come from depending upon such a highly specialized and restricted pool of insects. The ability to self provides some reproductive assurance [23]. In like manner, type I and choking type II(pop) *Epichloë* have no means to reproduce except through transfer of spermatia. If *Botanophila* flies fail in this regard, the fungi will not reproduce. Thus, sole dependence upon *Botanophila* might be expected to be relaxed, in favor of selection for spermatia transfer by multiple means, which could include ascosporic fertilization as well as non-*Botanophila* vectors.

## 4. Hypotheses

### 4.1. The Reproductive Assurance Hypothesis

(1) The degree of mutualism between *Botanophila* flies and *Epichloë* fungi depends upon the reproductive mode of the fungi. Greater specialization between partners should occur when the fungi can reproduce asexually without the services of the flies. When (asexual) reproduction is assured though invasion of host seeds, then *Epichloë* can “afford” to specialize on *Botanophila* flies as vectors of spermatia.

This hypothesis may seem counterintuitive in that it states fungi that only reproduce sexually should depend less on their primary vector, *Botanophila* flies. Why evolve toward less dependence if the fungi can only reproduce when spermatia are transferred? The answer is that less (rather than more) dependence should evolve if reproductive assurance is selected for in this system, as it apparently has been in the *Ophrys* orchid system (see above). If assurance of reproduction is not strongly selected for, then greater specialization between flies and fungi may indeed evolve, as articulated in the next hypothesis.

### 4.2. Vector Dependence

(2) A competing hypothesis is that greater specialization between flies and fungi will occur when *Epichloë* depends solely upon vectors of spermatia for its reproductive success. This hypothesis assumes that the “pollination” mutualism will coevolve toward greater and greater specialization between the two partners. Note that like hypothesis 1, this hypothesis also depends upon the reproductive mode of *Epichloë*, but its predictions are directly the opposite those from hypothesis (1); thus, properly designed empirical tests should be able to distinguish between these two competing hypotheses.

### 4.3. Predictions from Hypothesis (1)

*Botanophila* flies should be required for cross fertilization of type II(ind) *Epichloë,* but not for type I or type II(pop).

A.Type I and II(pop) *Epichloë* fungi should be visited by more species of *Botanophila* than type II(ind) fungi. Because type II(ind) *Epichloë* are hypothesized to be more specialized in their interaction, the interaction should be more species-specific than that with type I or type II(pop) species.B.Cross fertilization of type II(ind) *Epichloë* by *Botanophila* adults should enhance *Botanophila* larval development more than cross fertilization of type I or type II(pop) fungi.C. *Botanophila* visiting type II(ind) fungi should produce a higher proportion of nonviable eggs than those visiting type I or type II(pop) fungi. Because type II(ind) *Epichloë* risk less when interacting with *Botanophila*, they should be expected to minimize costs of fly larval feeding through increasing *Botanophila* mortality.D.Excluding slugs from stromata should reduce cross fertilization more in type I and II(pop) *Epichloë* compared to type II(ind) *Epichloë*.

Predictions from the Vector Dependence Hypothesis (2) would be directly opposite those above.

## 5. Future Directions

Predictions “A-E” should be amenable to experimental approaches. The first prediction (“A”) requires field data from type I, type II(ind), and type II(pop) *Epichloë* species. To date, published accounts exist only for two type II(ind) species (*E. elymi* in *Elymus* [16] and *E. festucae* in *Festuca rubra* [24]) (note: it is unclear if this was a type I or II(pop or ind) in the cultivated fields), one type II(pop) species (*E. typhina in P. distans* often fits this reproductive mode [15]), and one type I species (*E. typhina* in *D. glomerata—*from studies in the US [13] and Poland [14]). It is important that data be collected in such a way as to minimize the possibility of missing *Botanophila* visitation. Confirmation of visitation by flies cannot depend entirely on seeing deposited eggs on stromata, as flies may land, transfer spermatia, and then leave, without oviposition. It is also possible for eggs to fall off stromata, or for foraging ants to remove them. If one simply samples the stroma population at one point in time, it is not possible to avoid these pitfalls. Repeated observations of stromata, with careful written records, throughout the period of oviposition and perithecial development are required [25]. These observations should be combined with exclusion experiments in which stromata are bagged in mesh to prohibit flies from accessing them. Bags, of course, will also exclude other possible vectors, like slugs. It may be possible to exclude ascending slugs without excluding flies however, by using a band of adhesive gel (like Tanglefoot, Grand Rapids, MI) around culms below the stroma, as is commonly done to exclude ants from plant shoots [26].

Testing the prediction (“B”) that type I and II(pop) *Epichloë* fungi are visited by more species of *Botanophila* than are type II(ind) fungi will require collecting larvae or adult flies from stromata from a diversity of *Epichloë* species that represent all the reproductive modes. *Botanophila* taxonomy is based on adult males and because males are not found on stromata and larvae cannot be sexed, it is necessary to identify the flies through molecular means. This can be done using the cytochrome c oxidase II gene [27]. To date, there are data for four type I, two type II(pop), and six type II(ind) (Table 1). More sampling is needed; both of already sampled species to ensure we are not missing *Botanophila* species (some *Epichloë* species are represented by only a few samples), and of new species that have yet to be assessed (like, *E. typhina* in *P. autumnalis*, *Epichloe brachyelytri* in *Bachyelytrum erectum*, *Epichloe bromicola* in *Elymus tsukushiensis*, and *Epichloë sylvatica* ssp. pollinensis in *Hordelymus europaeus*).

Testing prediction “C” will require careful repeated sampling of marked *Epichloë* stromata in the field. Development of *Botanophila* eggs from deposition to hatch and subsequent larval development to pupation would need to be followed, as in Bultman [25]. Published data to date show that cross fertilization is required for fly development with *E. elymi* (type II[ind]) in *Elymus virginicus* [11,16]. In contrast, cross fertilization is not required for full development of larvae on the type I *E. typhina* in *Dactylis glomerata* [24].

Testing prediction “D” will require repeatedly visiting *Epichloë* stromata in the field to follow the type and fate of eggs deposited on stromata. Yellow/grey eggs are distinctly different from “normal” white eggs [29]. Doing this for several *Epichloë* species representing the three reproductive modes will be necessary. Now that DNA barcodes for the six European *Botanophila* species are available [33], eggs could be collected and their DNA extracted for molecular identification of *Botanophila* species. It should also be confirmed that yellow/grey eggs are, in fact, inviable by carefully following their development to determine if they ever hatch in the field (anecdotal evidence suggest they do not, pers. obs.).

One way *Epichloë* could affect *Botanophila* egg viability is through promoting infection of flies with *Wolbachia* bacterial parasites. *Wolbachia* is a sexual parasite of many arthropods and some nematodes [34]. It lives intracellularly within the reproductive tissues of its host and can skew the host sex ratio through male-killing as well as other effects on the host [34]. If *Wolbachia* led to inviable eggs through, for example male-killing, then stromata on which these male eggs were laid would not incur the larval feeding damage that would normally result if eggs had been viable. A mechanism by which the fungus could alter the infection status of flies is through production of antimicrobial agents that could disinfect flies of the bacterium. Interestingly, *Epichloë* are known to produce secondary compounds with antimicrobial properties (however, only antifungal, and not antibacterial, properties were tested) [35]. *Epichloë* that produce fewer antimicrobials would not impact *Wolbachia* infection and thus should incur more feeding damage by fly larvae. In contrast, *Epichloë* that produce high levels of antimicrobials could reduce *Wolbachia* infection which should lead to higher feeding damage of fungal reproductive propagules (ascospores). Under hypothesis 1, type I and type II(pop) *Epichloë* should produce low levels of antimicrobial compounds (and therefore not depress *Wolbachia,* which would lead to more inviable eggs and less larval feeding) because they depend less heavily on *Botanophila* for “pollination.” Therefore, the *Epichloë* would be expected to be weaker mutualists with *Botanophila*. The opposite would be true for type II(ind) *Epichloë*, which should be stronger mutualists and should therefore be less likely to harm *Botanophila*.

A possible test of this prediction could be done with *Drosophila*, which offers the advantages of having known infected lines and is easy to rear. Fruit flies infected with *Wolbachia* could be reared on artificial medium with or without *Epichloë* stromatal tissue added. Flies could then be assessed for *Wolbachia* infection.

A corollary of prediction “D” is that flies visiting type II(ind) should have higher infection rates of *Wolbachia* than those visiting type II(pop) or type I *Epichloë*. This could be tested by collecting many adults and/or larvae from stromata representing the two groups of reproductive systems and assessing *Wolbachia* infection. Some data of this sort exist see [30], but they are too limited to draw any firm conclusions.

While prediction “D” and its corollary flow from hypothesis 1, they are likely weak predictions, as even type II(ind) *Epichloë* should benefit from limiting *Botanophila* larval feeding—less feeding by the larvae should benefit reproductive output of these fungi as well (as long a male-killing does not substantially reduce the service of spermatia transfer by adult flies).

Prediction “E” could potentially be tested through applying adhesive gel (i.e., Tanglefoot, see above) barriers around culms below stromata. In this way slugs could be excluded from type I and II(pop) *Epichloë* stromata and perithecial development monitored in these compared to controls that lacked barriers. These results could be compared to similar experiments with type II(ind) *Epichloë.* Combinations of mesh bags and gel barriers could be used to evaluate the relative importance of *Botanophila* and slugs in cross fertilization.

## 6. Alternative Hypotheses

In this review we have considered an evolutionary hypothesis that depends upon the reproductive assurance of *Epichloë*. If there is low assurance due to lack of asexual reproductive capabilities, as in type I and II(pop), then *Epichloë* should not evolve toward a highly specialize interaction with *Botanophila*. It is possible that other selective pressures could be operating to produce the conflicting results found among *Epichloë-Botanophila* associations.

For example, type I and type II(pop) may not depend upon *Botanophila* for cross fertilization because their stromata are so numerous and concentrated in an area. This is certainly true for the cultivated commercial fields in Oregon, but less so for natural populations of native grasses in Poland, like *P. distans* infected with type II(pop) *E. typhina*. Yet, type I *Epichloë* typically occur in dense populations, with abundant stromata (pers. obs.). If an advantage of *Botanophila* to *Epichloë* is long distance dispersal, *Epichloë* living in dense clumps may not need the services of flies. *Botanophila* can cover long distances and often locate and lay eggs on very isolated stromata (pers. obs.). If stromata are not widely separated from one another, transfer of spermatia by flies should be less advantageous than for *Epichloë* that have widely isolated stromata. Less mobile agents, like slugs, may be adequate vectors for type I and type II(pop) *Epichloë*.

Yet, another alternative hypothesis has to do with resource concentration that type I and type II(pop) stromata provide adult flies. Female flies feed on stromatal tissue and presumably nothing else. So, this should be a very important food source for them. Intuitively, one would expect that a highly concentrated patch of stromata would attract more specialized individuals and species of pollinators, yet a test of this hypothesis with a native flowering shrub and its insect visitors in France found the opposite [36]. So, by inference, dense stands of type I and II(pop) stromata may actually attract more non-*Botanophila* visitors, like slugs, than the widely spaced stromata of type II(ind) *Epichloë.* These and possibly other alternative hypotheses are not mutually exclusive of the reproductive assurance hypothesis presented here. Careful experimental field studies will be required to distinguish between them.

## Figures and Tables

**Table 1 jof-08-01270-t001:** Host plant, *Epichloë* and *Botanophila* associations with respect to reproductive mode of *Epichloë. ** Individuals of these fungi have been found that are type I, type II, or type III.

Host	Fungus	Reproductive Mode	*Botanophila*	Location	Comment	Reference
*Poa trivialis*	*E. typhina*	I	*B. phrenione, B. dissecta, B. laterella*	Europe	Grass has woodland and open habitat varieties	[27]; unpubl. data
*Poa autumnalis*	*E. typhina poae*	II(ind)	?	Eastern US	Only one population with choke known, need fly	
*Poa nemoralis*	*E. typhina poae*	II(ind)	*B. dissecta*	Europe	Limited sampling	[27]
*Poa pratensis*	*E. typhina poae*	I	*B. dissecta, B. lobata*	Europe		[27,28]
*Dactylis glomerata*	*E. typhina*	I	*B. phrenione, B. lobata, B. dissecta*	Europe, Oregon (US)	US population of choke formers introduced	[27]
*Anthoxanthum odoratum*	*E. typhina*	I	*B. phrenione, B. dissecta*	Europe		[27]
*Brachypodium pinnatum*	*E. typhina*	I	*B. dissecta, B. phrenione, B. laterella*	Europe		[27]
*Holcus lanatus*	*E. typhina clarkii*	I	*B. dissecta, B. laterella, B. phrenione*	Europe		[27,29]
*Puccinellia distans*	*E. typhina*	II(ind) *	*B. dissecta, B. phrenione, B. cuspidata*	Poland	Appears to be type I within individuals	[30]
*Phleum pratense*	*E. typhina*	I	*B. dissecta, B. lobata*	Europe	Limited sampling	[27]
*Agrostis stolonifera, Agrostis tenuis*	*E. baconii*	I	*B. dissecta*	Europe	Limited sampling	[27,29]
*Festuca rubra*	*E. festucae*	II(ind)	*B. dissecta, B. lobata*	Europe, N. America		[27]
*Bachyelytrum erectum*	*E. brachyelytri*	II(ind)	?	N. America		
*Agrostis hyemalis, Sphenopholis obtusata*	*E. amarillans*	II(ind)	*B. lobata*, Taxon 5	Eastern US	Only taxon 5 on A. hyemalis	[27]; unpubl.
*Bromus erectus, Elymus repens*	*E. bromicola*	I	*B. dissecta, B. lobata*	Europe		[27,31]
*Bromus benekenii*	*E. bromicola*	II(ind?)	*B. lobata, B. laterella*	Europe	Limited sampling	[30]
*Elymus virginicus*	*E. elymi*	II(ind)	*B. lobata,* Taxon 5, Taxon 6	N. America		[27,30]
*Elymus canadensis*	*E. elymi*	II(ind)	Taxon 5	N. America		[30]
*Brachypodium sylvaticum*	*E. sylvatica*	II(ind) *	*B. lobata, B. phrenione, B. dissecta*	Europe		[27]
*Hordelymus europaeus*	*E. sylvatica pollinensis*	II(ind)	?	Europe		[32]
*Glyceria striata*	*E. glyceriae*	I	*Taxon 6*	Eastern US	Limited sampling	[27]

## Data Availability

Not applicable.

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
