# Peer review of "Does the Degree of Mutualism between Epichloë Fungi and Botanophila Flies Depend upon the Reproductive Mode of the Fungi?"

_jof, 2022, doi:10.3390/jof8121270_

Round 1
Reviewer 1 Report
Though such interaction is a very interesting, but you have raised more questions than answers. It would be more significant when it is more technically investigated.
Author Response
Dear Review #1:
Thank you for taking the time to review the manuscript. I agree that we have raised more questions than answers, but that is the point of a review article, which this is. For this reason, I made no major revisions in light of your review.
Reviewer 2 Report
In my opinion, the manuscript is interestingly written and contains important information. This is a good summary of the current knowledge on the topic presented. However, before a manuscript is accepted for publication, it requires minor linguistic / editorial corrections. Therefore, I am asking the authors to reread the work and remedy any shortcomings.
Author Response
Dear Review #2:
Thank you for reviewing the manuscript. I have reread the paper and made a few minor changes (in response to another reviewer), but found no linguistic errors.
Reviewer 3 Report
The paper comprehensively introduces the different kinds of interactions between Epichloë fungi and Botanophila flies. The previous mutualistic association between Epichloë fungi and Botanophila flies is now challenged by the fact that flies were not necessary for cross-fertilization of the fungus. Therefore, authors try to reconcile the current findings from an evolutionary perspective. Authors raise two competing hypotheses and corresponding predictions are given. Meanwhile, detail future directions and alternative hypotheses are also provided.
In general, the content of the manuscript is well organized and includes the current progress related to the interaction between Epichloë fungi and Botanophila flies. Based on the interpretation of the current findings, reasonable hypotheses and predictions are made.
In this review, there are many different types of Epichloë. To better understand the interactions and hypotheses and corresponding predictions, authors could include a Figure to illustrate two hypotheses and how different types of Epichloë are involved in these two hypotheses.
In the Abstract, only Type I and Type II are mentioned, but in the main text, Type III is introduced first. Please consider the order of the introduction.
In line 117-121, authors mention that “Type I (only sexual reproduction) should be less dependent on flies than Type II (sexual reproduction and asexual reproduction)” and explain “Members of the first group will incur a higher risk of reproductive failure due to their close dependence on Botanophila vectors”. Therefore, from my point of view, should Type I (only sexual reproduction) be MORE dependent on flies than Type II (sexual reproduction and asexual reproduction)?
Author Response
Thank you for taking the time to review the mansucript. If you look carefully at the abstract, all reproductive modes (I, II,, III) are described in the Abstract. And, in the same order as they are described in the Introduction. So, I have decided to keep the Introduction as is.
Your point about the Reproductive Hypothesis being counterintuitive, is well taken. I agree! Yet, the literature suggests this may not always be the trajectory taken as pollination systems evole. So, I will add the paragraph below at the end of the section on the Reproductive Assurance Hypothsis:
This hypothesis may seem counterintuitive in that it states fungi which only reproduce sexually should depend less on their primary vector, Botanophila flies. Why evolve toward less dependence if the fungi can only reproduce when spermatia are transferred? The answer is that less (rather than more) dependence should evolve if reproductive assurance is selected for in this system, as it apparently has been in the Ophrys orchid system (see above). If assurance of reproduction is not strongly selected for, then greater specialization between flies and fungi may indeed evolve, as articulated in the next hypothesis.
Round 2
Reviewer 1 Report
same comment as before
Author Response
Dear Review #1:
Your comments from before - "Though such interaction is a very interesting, but you have raised more questions than answers. It would be more significant when it is more technically investigated. "
I agree with this assessment in that it will be come more significant with more study, but as I said before, this is a review and the purpose is to review the current state of our knowledge, not to present new data.
I have added a paragragh to p. 8 that explains the initially counterintuitive hypothesis.
